# Guideline-Concordant Antibiotic Treatment for Hospitalised Patients with Community-Acquired Pneumonia and Clinical Outcomes at a Tertiary Hospital in Australia

**DOI:** 10.3390/antibiotics14080845

**Published:** 2025-08-20

**Authors:** Yogesh Sharma, Arduino A. Mangoni, Subodha Sumanadasa, Isuru Kariyawasam, Chris Horwood, Campbell Thompson

**Affiliations:** 1Department of Acute and General Medicine, Flinders Medical Centre, Adelaide, SA 5042, Australia; subodha.sumanadasa@sa.gov.au (S.S.); isuru.kariyawasam@sa.gov.au (I.K.); 2College of Medicine & Public Health, Flinders University, Adelaide, SA 5042, Australia; arduino.mangoni@flinders.edu.au; 3Clinical Improvement Unit, Flinders Medical Centre, Adelaide, SA 5042, Australia; chris.horwood@sa.gov.au; 4Discipline of Medicine, University of Adelaide, Adelaide, SA 5005, Australia; campbell.thompson@adeliade.edu.au

**Keywords:** community-acquired pneumonia, guideline-concordant antibiotic therapy, mortality, readmissions, hospital length of stay, intensive care unit admission

## Abstract

**Background/Objectives:** Community-acquired pneumonia (CAP) remains a major cause of hospitalisation and death, particularly among older and frail adults. Although treatment guidelines exist, adherence to empiric antibiotic recommendations is variable. This study examined whether receiving guideline-concordant antibiotics for CAP was associated with better short- and long-term clinical outcomes. **Methods:** We conducted a retrospective cohort study of adults admitted with radiologically confirmed CAP to a tertiary hospital in Australia from 1 January to 31 December 2023. Patients with hospital-acquired pneumonia or COVID-19 were excluded. Antibiotic concordance was assessed against local guidelines. Propensity score matching (PSM) accounted for 16 covariates including age, comorbidities (Charlson Index), frailty (Hospital Frailty Risk Score), and pneumonia severity (SMART-COP). Primary outcomes were in-hospital, 30-day, and one-year mortality. Secondary outcomes included ICU admission, invasive ventilation, vasopressor use, hospital length of stay, and 30-day readmissions. **Results:** Of 241 patients, 51.4% received guideline-concordant antibiotics. Mean age was 73.5 years; 50.2% were male; 42.2% had severe pneumonia (SMART-COP ≥ 5); 36.5% were frail. In unadjusted analysis, in-hospital mortality was higher in the concordant group (5.6% vs. 0.9%, *p* = 0.038). After PSM (n = 105 matched pairs), concordant treatment was associated with significantly lower 30-day mortality (coefficient = –0.12; 95% CI: –0.23 to –0.02; *p* = 0.018) and there was a non-significant trend towards reduced 1-year mortality (*p* = 0.058). Other outcomes, including in-hospital mortality, were not significantly different. **Conclusions:** Guideline-concordant antibiotics were associated with reduced 30-day mortality in CAP. These results support adherence to evidence-based treatment guidelines to improve patient outcomes.

## 1. Introduction

Community-acquired pneumonia (CAP) remains a leading cause of morbidity and mortality globally, particularly among older adults and immunocompromised individuals [1]. Among those hospitalised with CAP, short-term mortality ranges from 5% to 15% and can exceed 30% in patients requiring intensive care unit (ICU) admission [2,3].

Over the past three decades, the development and dissemination of evidence-based clinical practice guidelines have significantly advanced the management of CAP [4,5]. In Australia, empiric antibiotic recommendations for CAP differ slightly between jurisdictions but are primarily determined by disease severity. In South Australia, for patients with CAP, first-line antibiotic therapy is stratified according to disease severity. While traditional severity scores such as SMART-COP are acknowledged, current South Australian guidelines emphasise the use of clinical “red flags”—as outlined in the Australian Therapeutic Guidelines: Antibiotic—over older scoring systems [6]. These red flags are preferred because they align more closely with real-world clinical decision-making and help identify patients who can be safely managed on the ward versus those who may require ICU-level care. For low-severity CAP, the guideline recommends oral amoxycillin and/or doxycycline. For moderate-severity CAP, the preferred regimen is intravenous benzylpenicillin combined with oral or intravenous azithromycin. In patients with a confirmed severe penicillin allergy, respiratory fluoroquinolone is recommended as monotherapy. For patients with severe CAP requiring ICU admission, local guidelines advise the use of a broad-spectrum β-lactam in combination with either a macrolide or a fluoroquinolone, depending on allergy status [6].

Despite these recommendations, real-world adherence to guideline-concordant antibiotic therapy remains suboptimal. Costantini et al. [7] reported that only 56.7% of hospitalised CAP patients in Italy received guideline-concordant empiric therapy, although adherence was associated with reduced hospital length of stay (LOS) (mean reduction of 2.9 days) and antibiotic treatment duration (mean reduction of 2 days). An Australian study of 200 CAP patients found concordance rates of only 16% for initial antibiotic selection and 39% for subsequent therapy [8].

While numerous studies have demonstrated the effectiveness of guideline-recommended antibiotics in improving short-term outcomes, emerging evidence suggests that the impact of CAP may extend well beyond the acute phase. Survivors of CAP are at increased risk for long-term complications including functional decline, cardiovascular events, and death [9,10,11]. However, few studies have examined whether guideline-concordant therapy influences these longer-term outcomes. Notably, a recent study by Corrales-Medina et al. [10] reported a 50% reduction in cardiovascular mortality one year after CAP among older patients who received guideline-concordant antibiotics (HR 0.53; 95% CI 0.34–0.80; *p* = 0.003). However, this study was limited to patients aged ≥65 years and did not adjust for important confounders such as frailty or the use of cardioprotective medications (e.g., statins, antiplatelets, anticoagulants).

The present study evaluated whether receipt of guideline-concordant initial empiric antibiotic therapy for CAP is associated with improved short-term outcomes, including in-hospital and 30-day mortality, as well as reduced all-cause mortality at one year post-hospitalisation. Importantly, we accounted for key prognostic factors such as frailty—measured using the Hospital Frailty Risk Score (HFRS)—and the use of cardiovascular medications.

We hypothesised that adherence to guideline-recommended empiric antibiotic therapy was associated with both improved short-term clinical outcomes and lower long-term mortality among patients hospitalised with CAP.

## 2. Results

Over the study period, 510 patients with CAP were evaluated for guideline-concordant antibiotic treatment. Of these, 178 patients were excluded due to multiple admissions, and a further 91 patients were excluded for various other reasons, resulting in a final cohort of 241 patients (Figure 1). Among these, 124 patients (51.4%) received initial guideline-concordant treatment, while 117 (48.6%) did not.

The mean (SD) age of the cohort was 73.5 (17.9) years (range: 18–100 years), and 121 patients (50.2%) were male. Pneumonia severity at admission was documented by clinicians in 53 cases (21.9%). Among these, the SMART-COP score was the most commonly used tool (94.3%), followed by the CURB-65 score. Patients who had pneumonia severity assessed at the time of admission were similar in age, frailty status, and SMART-COP scores to those without a recorded severity score but had a lower comorbidity burden. Receipt of initial guideline-concordant antibiotic treatment did not differ between the two groups (*p* > 0.05).

The retrospectively calculated SMART-COP score was 2.9 (1.9). One hundred and forty-two patients (42.2%) were classified as having severe pneumonia based on the SMART-COP score (≥5). Among patients with prospectively recorded pneumonia severity scores at the time of hospital admission, agreement with retrospectively calculated SMART-COP severity was high, with 88.7% observed agreement. The Cohen’s kappa was 0.66 (*p* < 0.001), indicating substantial agreement beyond chance.

The mean (SD) HFRS was 4.6 (4.0), and 88 (36.5%) CAP patients were frail. The most commonly prescribed initial antibiotic regimen was a combination of a cephalosporin and a macrolide, used in 122 patients (50.6%), followed by penicillin plus a macrolide in 78 patients (32.3%). Use of ceftriaxone was significantly higher among patients who had severe CAP (87.8%) as determined by the SMART-COP score (≥5); however, 52% patients with mild-moderate CAP also received this antibiotic. The mean (SD) duration of initial intravenous antibiotic therapy was 3.6 (2.3) days, and the subsequent oral antibiotic treatment had a mean duration of 4.7 (2.2) days. Changes to antibiotic therapy following culture results or clinical response were not systematically recorded or analysed in this study, as the focus was on the initial empirical antibiotic treatment prescribed at hospital admission.

Patients who received guideline-concordant treatment were significantly younger and had higher pneumonia severity, as reflected by a greater mean SMART-COP score (*p* < 0.05). They also had significantly higher urea levels and were more likely to receive quinolones, although the number of patients receiving quinolones was small, warranting cautious interpretation of this finding (*p* < 0.05; Table 1). Other baseline characteristics were comparable between the two groups (*p* > 0.05; Table 1).

### 2.1. Clinical Outcomes Before PSM

In the unadjusted analysis, in-hospital mortality was significantly higher in the guideline-concordant treatment group compared to the non-concordant group (5.6% vs. 0.9%, *p* = 0.038). However, other clinical outcomes, including LOS and readmissions, did not differ significantly between the two groups (*p* > 0.05; Table 2).

### 2.2. Outcomes After PSM

PSM generated 105 matched pairs, achieving a SMD of less than 10% across matched covariates (Table 3 and Figure 2). Following matching:In-hospital mortality was not significantly different between groups (coefficient = −0.04; robust SE = 0.06; 95% CI: −0.16 to 0.07; *p* = 0.442).30-day mortality was significantly lower in the guideline-concordant group compared to the non-concordant group (coefficient = −0.12; robust SE = 0.05; 95% CI: −0.23 to −0.02; *p* = 0.018).1-year mortality showed a trend towards a reduction in the concordant group (coefficient = −0.11; robust SE = 0.06; 95% CI: −0.22 to −0.01; *p* = 0.058).

Other clinical outcomes, such as LOS and readmission rates, remained similar between the two groups (Table 4).

### 2.3. Sensitivity Analysis

Sensitivity analyses confirmed the robustness of the main findings. Among CAP patients who were aged ≥80 years, 30-day mortality was significantly lower in those who received guideline-concordant treatment (coefficient = −0.21; robust SE = 0.03; 95% CI: −0.29 to −0.13; *p* < 0.001). Similarly, among patients with higher pneumonia severity (SMART-COP score > 2), concordant treatment was associated with a significantly lower 30-day mortality (coefficient = −0.12; robust SE = 0.04; 95% CI: −0.20 to −0.04; *p* = 0.003).

## 3. Discussion

This study suggests that nearly half of patients hospitalised with CAP received initial guideline-concordant antibiotic treatment. These patients were generally younger, had more severe pneumonia, and were more likely to receive quinolones. In unadjusted analyses, in-hospital mortality was higher among those who received guideline-concordant therapy compared to those who did not, but other outcomes were similar between the two groups. However, after adjustment using PSM, there was no difference in in-hospital mortality between the two groups. Notably, patients who received guideline-concordant treatment had significantly lower 30-day mortality and showed a trend toward reduced 1-year mortality compared to those who received non-concordant treatment.

### 3.1. Adherence with Guideline-Directed Treatment

Our results suggest that a substantial proportion of patients with CAP did not receive guideline-concordant antibiotic treatment. Nevertheless, the rate of compliance observed in our study was somewhat higher than that reported in a recent Australian audit of 200 general medical inpatients, in which pneumonia severity was documented in only 9% of cases, and only 16% of patients received initial guideline-concordant therapy [8]. In contrast, our findings indicate lower compliance than a recent nationwide Australian hospital survey, which reported an overall 75% adherence to antibiotic-prescribing guidelines for CAP [12]. This discrepancy may reflect methodological differences, including the degree of detail sought during case note review and the use of stricter, locally developed prescribing guidelines as the reference standard in our study.

In our cohort, although pneumonia severity scores were documented by clinicians in only 21.9% of cases, we retrospectively calculated severity scores for all patients using available clinical data. This approach allowed for a comprehensive assessment of pneumonia severity in our analysis. However, the low rate of prospective documentation may still limit the utility of severity scores for guiding real-time clinical decision-making and may have contributed to variability in antibiotic prescribing, as guideline recommendations rely on severity stratification.

The most frequently prescribed antibiotic regimen in our cohort was a combination of a cephalosporin and a macrolide, aligning with findings from previous studies [8,13]. Consistent with these reports, our study also found that a substantial proportion (52%) of patients with mild to moderate CAP (SMART-COP < 5) were prescribed ceftriaxone. This may represent potential overtreatment, as current guidelines [5,14] recommend narrower-spectrum antibiotic therapy in such patients, and ceftriaxone is generally reserved for those with more severe disease. This pattern of overtreatment has been previously highlighted by Trad et al. [13], who reported that patients with mild CAP, as assessed by the Pneumonia Severity Index (PSI), were eight times more likely to receive treatment appropriate for severe disease compared to those whose management aligned with their severity classification (OR 8.2, 95% CI 1.7–40.3, *p* < 0.009).

### 3.2. Short Term Outcomes

#### 3.2.1. In-Hospital and 30-Day Mortality

Our study suggests that the use of guideline-concordant antibiotic treatment for CAP was associated with a significant reduction in short-term mortality, with a trend toward lower one-year mortality, although the latter did not reach statistical significance. These findings are broadly consistent with a Dutch study by Huijts et al. [15], which included 1047 patients with confirmed CAP and found that 62.9% received guideline-adherent antibiotics based on pneumonia severity determined by three different severity scores, including the PSI. That study reported a non-significant trend toward reduced in-hospital mortality (adjusted OR 0.77, 95% CI 0.37–1.61), similar to our findings, which showed a PSM trend toward reduced in-hospital mortality (coefficient −0.04, 95% CI −0.16 to 0.07). However, important methodological differences exist: The Dutch study adjusted only for pneumonia severity, while our analysis incorporated additional confounders, including comorbidity burden, frailty, and cardiovascular medication use. Notably, we found that 30-day mortality was significantly lower among patients who received guideline-concordant treatment, reinforcing the potential benefit of adherence to antibiotic guidelines.

Similarly, an Italian study by Rossio et al. [16], involving 191 CAP patients, also observed a trend toward lower in-hospital mortality among those treated according to clinical guidelines (OR 0.66, 95% CI 0.25–1.79). However, there was no significant difference in three-month mortality after adjustment for age, gender, treatment adherence, and pneumonia severity. A key distinction between our study and these previous studies is the broader adjustment for clinically relevant factors such as frailty and comorbidities, which may influence treatment outcomes. The observed improvement in short-term outcomes with guideline-concordant antibiotic therapy may reflect its efficacy in targeting the most common pathogens responsible for CAP, thereby reducing the risk of treatment failure associated with empiric regimens that are not aligned with established guidelines [17].

#### 3.2.2. LOS

Previous research [7] has reported a reduction in LOS with the use of guideline-concordant antibiotic treatment for CAP. However, this association was not observed in our study. A possible explanation is the higher clinical complexity of our cohort, which included an older population (mean age 73.5 vs. 72.6 years) and a greater burden of comorbidities (severe comorbidity in 17.4% vs. 14.8%) and frailty. In such patients, appropriate antibiotic selection may have a relatively smaller influence on LOS and in-hospital mortality compared to younger, less complex individuals. Previous research suggests that the clinical trajectory in frail older adults with multimorbidity and polypharmacy is often shaped by factors beyond the infectious process alone, including baseline functional status, medication interactions, and delayed recovery and its effects on deconditioning the patient [18,19]. These factors may attenuate the expected benefits of optimal antibiotic therapy on hospital-related outcomes in this population.

### 3.3. One-Year Mortality

Our study observed a trend toward reduced one-year mortality among patients who received guideline-concordant antibiotic treatment for community-acquired pneumonia (CAP). This finding is consistent with a Canadian cohort study involving 1909 patients, which similarly reported a trend toward lower all-cause mortality at one year (hazard ratio [HR] 0.82; 95% CI 0.65–1.04; *p* = 0.099) [10]. Notably, that study also found a significant 50% reduction in cardiovascular mortality one year post-CAP admission (HR 0.53; 95% CI 0.34–0.80; *p* = 0.003).

However, key differences exist between the two studies. The Canadian study was limited to patients aged ≥65 years and did not account for important confounding variables such as frailty or the use of cardioprotective medications (e.g., statins, antiplatelets, anticoagulants). In contrast, our study adjusted for a broader range of clinical variables, providing a more nuanced assessment of the relationship between guideline adherence and long-term outcomes.

It is plausible that failure to administer guideline-concordant antibiotics may lead to greater disease severity, prolonged inflammation, and higher cardiovascular risk following CAP. Persistent systemic inflammation has been implicated in the pathogenesis of post-CAP cardiovascular complications, including myocardial infarction, heart failure, and stroke [20,21]. Conversely, antibiotics included in guideline-concordant regimens—particularly macrolides, doxycycline, and fluoroquinolones—may confer additional benefit through their immunomodulatory effects, thereby mitigating inflammation-driven cardiovascular events [22,23,24]. These findings highlight a potential long-term benefit of guideline-concordant antibiotic therapy that extends beyond the acute infection period, particularly in reducing cardiovascular risk after CAP.

### 3.4. Limitations

This study has several important limitations. First, we were unable to determine the time from hospital admission to antibiotic administration—a factor known to influence clinical outcomes in CAP, particularly in severe cases. Second, although propensity score matching (PSM) was used to control for multiple confounding variables, the possibility of residual or unmeasured confounding remains. As such, the observed associations should be interpreted with caution. Third, the study did not account for pre-hospital factors, such as prior healthcare interventions, medication adherence, vaccination status, or health-related behaviours (e.g., smoking, physical activity, or nutritional status), all of which may have influenced both the severity of illness at presentation and longer-term outcomes, including mortality. This study focused solely on outcomes associated with initial empiric antibiotic selection and did not assess concordance for subsequent oral or step-down antibiotic therapy. Additionally, changes or escalation of antibiotic treatment during hospitalisation were not captured, which may have influenced clinical outcomes and affected the assessment of true guideline adherence throughout the course of care. Fourth, while our study observed a trend toward reduced 1-year mortality associated with guideline-concordant antibiotic therapy, cause-specific mortality data were not available. Therefore, we cannot determine the extent to which cardiovascular versus non-cardiovascular causes contributed to long-term mortality outcomes. This limitation warrants cautious interpretation of the association between initial antibiotic treatment and long-term survival.

## 4. Materials and Methods

This retrospective cohort study was conducted at Flinders Medical Centre, a tertiary hospital in South Australia serving a population of approximately 450,000 in Adelaide’s southern suburbs. Adult patients (≥18 years) admitted with CAP between 1 January 2023 and 31 December 2023 were identified using International Classification of Diseases 10th Revision Australian Modification (ICD-10-AM) codes J12–J18.9 [25]. CAP was defined by the presence of fever, cough (with or without sputum), and/or dyspnoea in conjunction with radiological evidence of pulmonary infiltrates [26]. Patients were excluded if they had hospital-acquired pneumonia (HAP) (onset ≥48 h after admission) or tested positive for SARS-CoV-2 on reverse transcription polymerase chain reaction (RT-PCR) during their hospital stay. Ethics approval for the study was obtained from the Southern Adelaide Local Health Network Human Research Ethics Committee (approval number: LNR/23/SAC/231 on 8 December 2023).

Clinical and demographic data were extracted from structured reviews of case notes and electronic medical records (EMR). A panel of senior clinicians determined whether patients received initial empiric antibiotic treatment concordant with the Southern Adelaide Local Health Network CAP guidelines, the local antibiotic policy in place during the study period, and whether pneumonia severity scoring was documented at the time of admission. Pneumonia severity was also assessed retrospectively using SMART-COP score. The SMART-COP score [27] was used to evaluate the need for intensive respiratory or vasopressor support. This scoring was determined from the following parameters: systolic blood pressure, multilobar infiltrates, albumin, respiratory rate, tachycardia, oxygenation, and pH, with a maximum possible score of 11 and higher score indicative of greater severity of pneumonia [14].

Microbiological aetiology was determined using culture and respiratory viral RT-PCR results. Antibiotics administered during admission were determined from the hospital pharmacy database. We also determined whether patients were receiving statins, antiplatelets, and anticoagulants prior to their admission. The comorbidity burden was determined using the Charlson comorbidity index (CCI) [28]. Frailty status was assessed using the Hospital Frailty Risk Score (HFRS), and patients with HFRS ≥5 were classified as frail [29]. Furthermore, routine blood investigations on admission, such as haemoglobin, white blood cell (WBC) count, C-reactive protein (CRP), albumin, creatinine, and international normalised ratio (INR), were extracted from the EMR.

### 4.1. Clinical Outcomes

The primary short-term outcomes included in-hospital mortality and 30-day all-cause mortality from the date of admission. The long-term outcome was all-cause mortality within one year of the index hospitalisation. Secondary outcomes included admission to the ICU, need for invasive mechanical ventilation or vasopressor support, LOS, and 30-day unplanned readmissions within 30 days of discharge.

### 4.2. Statistical Analysis

Continuous variables were tested for normality using the Shapiro–Wilk test and analysed with *t*-tests or Wilcoxon rank-sum tests, as appropriate. Categorical variables were compared using chi-square or Fisher’s exact tests.

### 4.3. Propensity Score Matching (PSM)

To address potential confounding in the comparison between patients who received initial guideline-concordant antibiotic treatment and those who did not, we applied PSM. A multivariable logistic regression model was constructed to estimate the probability of receiving guideline-concordant therapy, including covariates with *p*-values of <0.20 in univariate analyses. The final model included 16 covariates: age, sex, CCI, SMART-COP score, HFRS, microbiological aetiology, smoking status, history of alcoholism, haemoglobin, CRP, albumin, creatinine, microbiological aetiology, and use of statins, antiplatelet agents, and anticoagulants.

Matching was performed in a 1:1 ratio using nearest-neighbour matching without replacement [30]. Covariate balance between the matched cohorts was assessed using standardised mean differences (SMD), with an SMD >10% indicating meaningful imbalance [31]. Propensity score distributions were visually inspected before and after matching to confirm appropriate overlap. Outcomes between matched groups were compared using the average treatment effect on the treated (ATET). The results were reported as coefficients with robust standard errors (SEs) and 95% confidence intervals (CIs).

### 4.4. Sensitivity Analysis

Sensitivity analyses were performed in two subgroups: patients aged ≥ 80 years and those with moderate or severe pneumonia (SMART-COP > 2). A two-sided *p*-value of <0.05 was considered statistically significant across all analyses. All statistical analyses were performed using Stata software version 19.0 (StataCorp LLC, College Station, TX, USA).

## 5. Conclusions

In conclusion, this study suggests that the use of guideline-concordant antibiotic therapy in hospitalised patients with CAP is associated with improved short-term mortality and a trend toward reduced long-term mortality. These findings reinforce the importance of adhering to established clinical practice guidelines for the treatment of CAP and support continued efforts to improve compliance with recommended antibiotic regimens in routine clinical care [5,14]. However, given the absence of cause-specific mortality data, further research is needed to elucidate the mechanisms underlying the observed trends in long-term mortality.

## Figures and Tables

**Figure 1 antibiotics-14-00845-f001:**
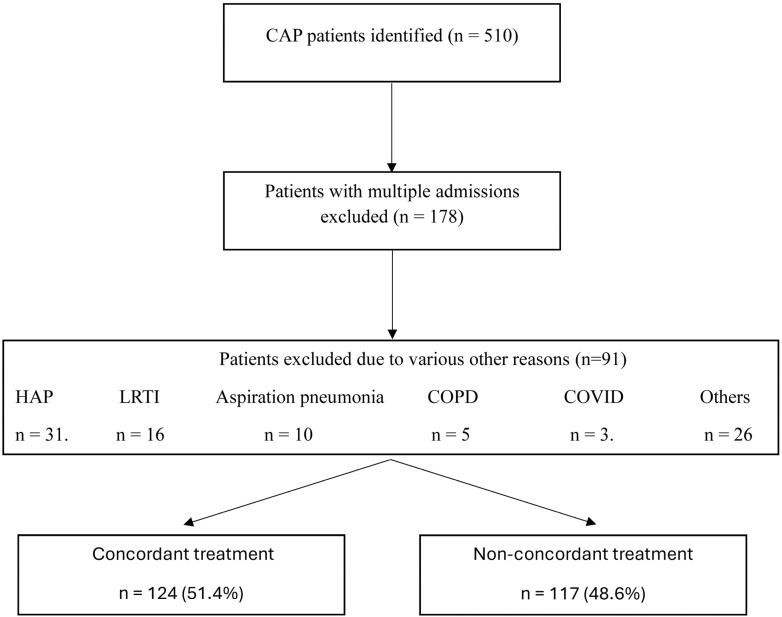
Study flow diagram.

**Figure 2 antibiotics-14-00845-f002:**
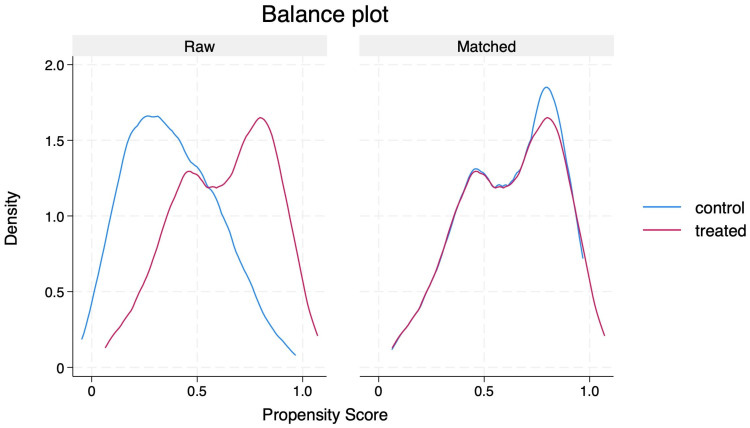
Density plot before and after propensity score matching.

**Table 1 antibiotics-14-00845-t001:** Characteristics of patients who received compared to those who did not receive guideline-concordant treatment for CAP.

	Total	Received Guideline-Concordant Treatment	Not Received Guideline-Concordant Treatment	*p*-Value
N (%)	241	124 (51.4)	117 (48.6)	
Age years mean (SD)	73.5 (17.9)	70.3 (20.1)	76.6 (15.0)	0.006
Sex male n (%)	121 (50.2)	58 (47.9)	63 (52.10)	0.272
Charlson index median (IQR)	1 (0, 4)	1 (0, 4)	1 (0, 3)	0.129
SMART-COP score mean (SD)	2.9 (1.9)	3.7 (1.9)	2.3 (1.7)	<0.001
HFRS mean (SD)	4.6 (4.0)	4.7 (4.1)	4.5 (4.1)	0.754
Frail	88 (36.5)	46 (37.1)	42 (35.9)	0.847
Current Smokers n (%)	16 (6.6)	8 (6.4)	8 (6.8)	0.904
Alcoholism * n (%)	7 (2.9)	6 (4.8)	1 (0.8)	0.066
Chronic lung disease n (%)	93 (38.6)	44 (37.6)	49 (39.5)	0.761
Coronary artery disease n (%)	28 (11.6)	21 (16.9)	7 (5.9)	0.08
Haemoglobin mean (SD)	123.3 (20.7)	122.2 (21.7)	124.5 (19.6)	0.387
WBC count mean (SD)	12.7 (6.2)	12.4 (5.9)	13.1 (6.3)	0.445
CRP median (IQR)	90.3 (32.4, 71.3)	93.7 (31.2, 169.3)	97.3 (41.6, 192.2)	0.376
Urea mean (SD)	8.2 (5.5)	9.0 (6.1)	7.3 (4.8)	0.015
Creatinine mean (SD)	97.7 (63.8)	104.5 (74.7)	90.5 (48.7)	0.090
INR mean (SD)	1.4 (0.6)	1.4 (0.7)	1.3 (0.4)	0.236
Microbiologic aetiology n (%)				0.951
No pathogen detected	189 (78.4)	96 (77.4)	93 (79.5)	
Bacterial	20 (8.3)	10 (8.1)	10 (8.5)	
Respiratory viruses	25 (10.4)	14 (11.3)	11(8.6)	
Polymicrobial	7 (2.9)	4 (3.2)	3 (2.5)	
Antibiotics n (%)				
Penicillin	94 (42.7)	45 (39.5)	49 (46.2)	0.312
Cephalosporins	133 (59.9)	67 (50.3)	66 (61.8)	0.603
Macrolides	199 (85.8)	101 (84.8)	98 (86.3)	0.687
Doxycycline	22 (10.1)	14 (12.2)	8 (7.8)	0.282
Quinolones	9 (4.3)	8 (7.3)	1 (1.0)	0.027
Others	6 (3.2)	2 (1.9)	4 (4.5)	0.316
Total duration of antibiotics in days, mean (SD)	7.7 (2.9)	7.6 (3.1)	7.7 (2.8)	0.796
Statins n (%)	100 (44.8)	55 (48.3)	45 (41.3)	0.296
Antiplatelets n (%)	60 (27.0)	29 (25.4)	31 (28.7)	0.584
Anticoagulants n (%)	78 (35.3)	44 (38.9)	34 (31.5)	0.246

* Alcoholism defined as a documented history of alcohol use disorder, alcohol dependence, or harmful alcohol use recorded in medical notes or coded diagnoses. CAP, community-acquired pneumonia; SD, standard deviation; IQR, interquartile range; SMART-COP, systolic blood pressure, multilobar infiltrates, albumin, respiratory rate, tachycardia, confusion, oxygen, and pH; HFRS, Hospital Frailty Risk Score; WBC, white blood cell; CRP, C-reactive protein; INR, international normalised ratio.

**Table 2 antibiotics-14-00845-t002:** Clinical outcomes according to guideline-concordant treatment.

Outcome	Received Guideline-Concordant Treatment	Not Received Guideline-Concordant Treatment	*p*-Value
LOS median (IQR)	3.7 (1.9, 7)	3.7 (1.8, 6.3)	0.599
ICU admission n (%)	8 (6.4)	4 (3.4)	0.279
Invasive ventilation	0	1 (0.9)	0.302
Vasopressor support n (%)	3 (2.4)	3 (2.5)	0.943
In-hospital mortality n (%)	7 (5.6)	1 (0.9)	0.038
Mortality within 30 days of admission n (%)	13 (10.4)	5 (5.1)	0.123
Mortality within 365 of admission n (%)	18 (14.5)	12 (10.3)	0.317
30-day readmission n (%)	14 (11.3)	23 (19.6)	0.072

LOS, length of hospital stay; ICU, intensive care unit.

**Table 3 antibiotics-14-00845-t003:** Standardised Mean Differences and Variance Ratios Before and After Matching.

Variable	SMD (Raw)	SMD (Matched)	Variance Ratio (Raw)	Variance Ratio (Matched)
Age	0.393	−0.05	0.536	0.828
Sex	−0.122	0.060	1.006	1.220
Charlson Index	0.101	−0.028	1.323	1.325
SMART-COP	0.818	0.000	1.468	1.078
HFRS	0.080	−0.012	1.133	1.094
Statin use	0.125	0.055	1.025	1.007
Antiplatelet use	−0.059	−0.057	0.944	0.826
Anticoagulant use	0.156	0.076	1.102	1.042
Haemoglobin	0.056	−0.187	1.032	0.945
CRP	−0.116	−0.039	1.018	1.059
Albumin	−0.056	−0.010	0.985	1.149
Urea	0.400	0.085	2.850	1.338
Creatinine	0.268	0.084	4.325	2.159
Microbiological aetiology	0.055	−0.109	1.117	0.859
Smoking	−0.008	0.000	0.974	1.000
Alcoholism *	0.224	0.031	4.679	1.024

* Alcoholism defined as a documented history of alcohol use disorder, alcohol dependence, or harmful alcohol use recorded in medical notes or coded diagnoses. SMD, standardised mean difference; SMART-COP, systolic blood pressure, multilobar infiltrates, albumin, respiratory rate, tachycardia, confusion, oxygen, and pH; HFRS, Hospital Frailty Risk Score; CRP, C-reactive protein.

**Table 4 antibiotics-14-00845-t004:** Clinical Outcomes After Propensity Score Matching in Patients Receiving Guideline-Concordant Treatment vs. Non-Concordant Treatment.

Outcome	ATET (Effect Estimate)	Robust SE	95% CI	*p*-Value
LOS	0.95	1.05	–1.11 to 3.30	0.364
ICU admission	0.03	0.03	–0.02 to 0.09	0.234
Vasopressor support	−0.01	0.02	–0.06 to 0.04	0.739
In-hospital mortality	−0.04	0.06	–0.16 to 0.07	0.442
Mortality within 30 days of admission	−0.12	0.05	–0.23 to –0.02	0.018
Mortality within 365 days of admission	−0.11	0.06	–0.22 to 0.01	0.058
30-day readmission rate	0.05	0.04	–0.03 to 0.12	0.252

ATET, average treatment effect on the treated; SE, standard error; CI, confidence interval; LOS, length of hospital stay; ICU, intensive care unit.

## Data Availability

Data available from corresponding author on reasonable request and after approval by the ethics committee.

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
