# Peer review of "Guideline-Concordant Antibiotic Treatment for Hospitalised Patients with Community-Acquired Pneumonia and Clinical Outcomes at a Tertiary Hospital in Australia"

_antibiotics, 2025, doi:10.3390/antibiotics14080845_

Round 1

Reviewer 1 Report

Comments and Suggestions for Authors

The authors have written a good conceptualise but I think two points need to be clarified 

  1. Was the antibiotic prescription(not concordant with guidelines) based on local antibiotic policy?
  2. Was any change made after AST and culture came positive? 
  3. In cases which were identified with Respiratory viruses, were antibiotics not discontinued? And if not, what was the reason for continuing antibiotics? 
  4. What was the method used for the detection of viruses in the case study ? 

Author Response

The authors have written a good conceptualise but I think two points need to be clarified 

  1. Was the antibiotic prescription(not concordant with guidelines) based on local antibiotic policy?

Response: “Guideline-concordant” antibiotic therapy was assessed against the local antibiotic policy during the study period, represented by the Southern Adelaide Local Health Network Community-Acquired Pneumonia guidelines. Prescriptions classified as “not concordant with guidelines” were therefore also not in accordance with the local antibiotic policy. We have now clarified this in the methods section on page 3.

“A panel of senior clinicians determined whether patients received initial empiric antibiotic treatment concordant with the Southern Adelaide Local Health Network CAP guidelines, the local antibiotic policy in place during the study period, and whether pneumonia severity scoring was documented at the time of admission.”

  1. Was any change made after AST and culture came positive? 

Response: In our study, we assessed concordance based on the initial empiric antibiotic prescription at the time of admission. While antibiotic regimens may have been modified after antimicrobial susceptibility testing (AST) and culture results became available, these changes were outside the scope of the present analysis and therefore were not reported.

  1. In cases which were identified with Respiratory viruses, were antibiotics not discontinued? And if not, what was the reason for continuing antibiotics? 

Response: Our study did not specifically identify or analyse cases where antibiotics were discontinued following detection of a respiratory virus. It is possible that in some patients, antibiotics were stopped, while in others they were continued based on the treating clinician’s assessment of possible concurrent bacterial pneumonia.

  1. What was the method used for the detection of viruses in the case study? 

Response: Respiratory viruses were detected using respiratory viral RT-PCR, as stated in the Methods section (page 3).

“Microbiological aetiology was determined using culture and respiratory viral RT-PCR results.”

Reviewer 2 Report

Comments and Suggestions for Authors

This is a well written manuscript, describing a single-center retrospective study which aimed to evaluate whether guideline-concordant empirical treatment of CAP affects short- and long-term outcomes in a propensity-matched cohort of patients admitted with CAP. Methodology is sound and well described, results are well documented and discussion is complete. There are several limitations across the study which the authors acknowledge. Conclusions are relevant to the findings.

I have some comments below:

Line 164: the fact that pneumonia severity was recorded in only 23% should be emphasized. Please also comment how this limits the predictive value of this factor.

Line 173: “Among patients with prospectively recorded pneumonia severity scores at admission” please define period that scores were recorded prospectively.

Lines 177-184: please clarify whether this information on antibiotic treatment refers to initial empirical treatment.

Please also mention whether changes in empirical treatment were recorded among the cohort.

Line 187 “They also had significantly higher urea levels and were more likely to receive quinolones” although patient numbers in this group are extremely low.

Lines 328-350: I understand the conclusion of this finding, but 1-year mortality can be attributed to other than cardiovascular factors. I believe the authors should mention this, and also acknowledge that their study recorded 1-year mortality without recording causes of death.

Author Response

This is a well written manuscript, describing a single-center retrospective study which aimed to evaluate whether guideline-concordant empirical treatment of CAP affects short- and long-term outcomes in a propensity-matched cohort of patients admitted with CAP. Methodology is sound and well described, results are well documented and discussion is complete. There are several limitations across the study which the authors acknowledge. Conclusions are relevant to the findings.

I have some comments below:

Line 164: the fact that pneumonia severity was recorded in only 23% should be emphasized. Please also comment how this limits the predictive value of this factor.

Response: Thank you for this important observation. While pneumonia severity scores were documented prospectively by clinicians in only 21.9% of cases, we retrospectively determined severity scores for all patients using clinical data extracted from medical records. This allowed a comprehensive assessment of pneumonia severity in our analyses. However, the low rate of prospective documentation may have limited the use of severity scoring to guide real-time clinical decision-making and potentially contributed to variability in antibiotic prescribing, since guideline recommendations for CAP treatment are stratified by severity. We have now emphasised this point in the Discussion section to clarify the potential impact of incomplete clinician documentation on treatment adherence and outcome prediction (page 11).

“In our cohort, although pneumonia severity scores were documented by clinicians in only 21.9% of cases, we retrospectively calculated severity scores for all patients using available clinical data. This approach allowed a comprehensive assessment of pneumonia severity in our analysis. However, the low rate of prospective documentation may still limit the utility of severity scores for guiding real-time clinical decision-making and may have contributed to variability in antibiotic prescribing, as guideline recommendations rely on severity stratification.”

Line 173: “Among patients with prospectively recorded pneumonia severity scores at admission” please define period that scores were recorded prospectively.

Response: Pneumonia severity scores were prospectively documented by clinicians at the time of hospital admission as part of routine clinical assessment throughout the study period. We have now clarified this period in the results section (page 6).

“Among patients with prospectively recorded pneumonia severity scores at the time of hospital admission...”

Lines 177-184: please clarify whether this information on antibiotic treatment refers to initial empirical treatment.

Please also mention whether changes in empirical treatment were recorded among the cohort.

Response: The information presented in Lines 177-184 refers specifically to the initial empirical antibiotic treatment prescribed at hospital admission. The mean duration reported reflects the initial intravenous antibiotic phase followed by oral therapy. Regarding changes to empirical treatment after initiation, these were not systematically recorded or analysed in this study, as our focus was on initial antibiotic selection and its association with clinical outcomes. We have now clarified this in the results section (page 6).

“The most commonly prescribed initial antibiotic regimen was a combination of a cephalosporin and a macrolide, used in 122 patients (50.6%), followed by penicillin plus a macrolide in 78 patients (32.3%).”

“Changes to antibiotic therapy following culture results or clinical response were not systematically recorded or analysed in this study, as the focus was on the initial empirical antibiotic treatment prescribed at hospital admission.”

Line 187 “They also had significantly higher urea levels and were more likely to receive quinolones” although patient numbers in this group are extremely low.

Response: Thank you for your observation. We acknowledge that the subgroup of patients receiving quinolones was relatively small. However, the difference in quinolone use between those receiving guideline-concordant versus non-concordant treatment reached statistical significance (P < 0.05), as presented in Table 1. We have now added a clarifying statement in the Results section (page 6) to highlight the small numbers and the need for cautious interpretation of this finding.

“They also had significantly higher urea levels and were more likely to receive quinolones, although the number of patients receiving quinolones was small, warranting cautious interpretation of this finding (P<0.05; Table 1).”

Lines 328-350: I understand the conclusion of this finding, but 1-year mortality can be attributed to other than cardiovascular factors. I believe the authors should mention this, and also acknowledge that their study recorded 1-year mortality without recording causes of death.

Response: Thank you for this important comment. We acknowledge that 1-year mortality can be influenced by a variety of factors beyond cardiovascular causes, and that our study did not capture cause-specific mortality data. We have now explicitly acknowledged this limitation in the revised manuscript to ensure cautious interpretation of the association between guideline-concordant antibiotic therapy and long-term mortality outcomes. Please refer to Limitations and Conclusions section (page 14).

“Fourth, while our study observed a trend toward reduced 1-year mortality associated with guideline-concordant antibiotic therapy, cause-specific mortality data were not available. Therefore, we cannot determine the extent to which cardiovascular versus non-cardiovascular causes contributed to long-term mortality outcomes. This limitation warrants cautious interpretation of the association between initial antibiotic treatment and long-term survival.”

“However, given the absence of cause-specific mortality data, further research is needed to elucidate the mechanisms underlying the observed trends in long-term mortality.”